# Acute Effects of Hesperidin in Oxidant/Antioxidant State Markers and Performance in Amateur Cyclists

**DOI:** 10.3390/nu11081898

**Published:** 2019-08-14

**Authors:** Francisco Javier Martínez-Noguera, Cristian Marín-Pagán, Jorge Carlos-Vivas, Jacobo A. Rubio-Arias, Pedro E. Alcaraz

**Affiliations:** 1Research Center for High Performance Sport, Catholic University of Murcia, 30107 Murcia, Spain; 2Faculty of Sport, Catholic University of Murcia, 30107 Murcia, Spain; 3Faculty of Sport, Research Center for High Performance Sport, Catholic University of Murcia Catholic University of Murcia, 30107 Murcia, Spain

**Keywords:** hesperidin, hesperitin, antioxidants, catalase, superoxide dismutase, reduced glutathione, oxidized glutathione, performance and exercise

## Abstract

Human and animal studies have shown that Hesperidin has the ability to modulate antioxidant and inflammatory state and to improve aerobic performance. The main objective of this study was to assess whether the acute intake of 500 mg of 2S-Hesperidin (Cardiose^®^) improves antioxidant status, metabolism, and athletic performance, during and after a rectangular test (aerobic and anaerobic effort). For this, a crossover design was used in 15 cyclists (>1 year of training), with one week of washout between placebo and Cardiose^®^ supplementation. After the intervention, significant differences in average power (+2.27%, *p* = 0.023), maximum speed (+3.23%, *p* = 0.043) and total energy (∑ 4 sprint test) (+2.64%, *p* = 0.028) between Cardiose^®^ and placebo were found in the best data of the repeated sprint test. Small changes were also observed in the activity of catalase, superoxide dismutase, reduced glutathione concentration and oxidized/reduced glutathione (GSSG/GSH) ratio, as well as the lipoperoxidation products (thiobarbituric acid reactive substances; TBARS), at different points of the rectangular test, although not significant. Our findings showed improvements in anaerobic performance after Cardiose^®^ intake, but not in placebo, suggesting the potential benefits of using Cardiose^®^ in sports with a high anaerobic component.

## 1. Introduction

The use of ergogenic aids in sports has increased considerably in recent years [1]. This growing interest is driven by the emergence of studies that have shown how ergogenic aid intake can contribute to improvement in athletic performance, post-exercise recovery, or antioxidant capacity enhancement, as well as changes in body composition (e.g., body fat loss or increase in lean muscle mass), by a stimulation of fatty acids mobilization [2,3,4,5,6]. Within the ergogenic aids, there exist several categories based on the degrees of evidence shown in scientific literature. For instance, category A refers to the highest level of evidence and includes proteins, amino acids, creatine, beta-alanine, carbohydrates, etc. [7]

One potentially promising group that can serve as an ergogenic aid is polyphenols [8]. Polyphenols are bioactive compounds which are widely distributed in plant and plant-based foods, such as vegetables, fruit, cocoa, tea, coffee, and wine [9]. Polyphenols are a very diverse group of compounds, with over 500 different molecules identified in foods, which can be divided into four main categories, according to their chemical structure: flavonoids (e.g., hesperidin, hesperitin, etc.), phenolic acids, stilbenes, and lignans [10]. In addition, polyphenols are of the most studied compounds for their positive effects on human health [11]. Specifically, these products are often used for chronic disease, delaying the ageing process, improving body composition, and increasing life expectancy. Moreover, polyphenols have been proposed to be beneficial in exercise and exercise performance [12,13,14,15]. In fact, some polyphenols, such as quercetin [8,16,17,18] or cocoa flavanols [19,20,21] have been extensively used for this purpose.

For example, polyphenols have been proposed to improve performance by increasing mitochondrial biogenesis in two different ways. Firstly, they stimulate stress-related cell signalling pathways that increase the expression of genes encoding cytoprotective proteins, such as nuclear respiratory factor 2 (NRF2) [22]. Secondly, it has been reported that these phytomolecules may modulate muscle function and mitochondrial biogenesis by activating sirtuins (SIRT1) and increasing PGC-1α activity. [23,24,25]. Moreover, polyphenols have been shown to work effectively against exercise induced oxidative stress [26,27], since as seen in many investigations, exercise increases reactive oxygen species (ROS) production, which may result in oxidative stress, and lead to muscle fibre damage, which eventually results in muscle fatigue [28,29,30,31]. Within the large family of biomolecules that are polyphenols, the most studied in the field of sports performance is quercetin [16,17], although new molecules of this family such as luteolin, mangiferin [32], and hesperidin [33] are being investigated.

Hesperidin is a polyphenol, specifically a flavonoid, that is mostly found in citrus fruits [34]. Hesperidin is the most relevant flavonoid in some citrus species, such as sweet orange (*Citrus sinensis*), finding high concentration of hesperidin in orange juice (up to 513 mg/L) [35]. Hesperidin is a chiral molecule, and can be found in two isomeric forms, as 2S- and 2R-Hesperidin. However, the 2S-Hesperidin form is naturally predominant in citrus fruits [36], being present in fresh sweet orange juice with an 2S isomer/2R isomer ratio of 15.4:1 in favour of the 2S-epimer (92% 2S-Hesperidin) [37]. However, during the industrial extraction and isolation of Hesperidin, part of this 2S-epimer naturally present in the citrus fruits is transformed into the 2R-epimer. In commercial hesperidin samples, the 2S isomer/2R isomer ratio is close to 1.5:1 in favour of the 2S-epimer (about 60% 2S-Hesperidin). Cardiose^®^ is a natural orange extract, produced by HealthTech BioActives (Murcia, Spain), that due to its unique manufacturing process maintains most of the natural hesperidin isomeric form (NLT 85% 2S-Hesperidin).

Hesperidin antioxidant [38], anti-inflammatory [39], and health promoting [40] properties have been extensively described. Moreover, the intake of hesperidin improves the nitric oxide (NO) synthesis through the activation of phosphorylation of proto-oncogene tyrosine-protein kinase (Src), protein kinase B (Akt), adenosine monophosphate-activated protein kinase (AMPK), and endothelial NO synthase, leading to an increased flow-mediated dilation [41]. An increased NO production leads to an improved endothelial function, allowing an enhanced O_2_ transport to working muscles during acute exercise and prolonged exercise [42]. In humans, the intake of flavanone-rich foods (a type of flavonoids including Hesperidin) has been linked to an increase in NO production, increased endothelium-dependent microvascular reactivity, as well as a reduction in diastolic blood pressure [43,44,45,46,47]. Endothelial function and different cardiovascular parameters were also improved after hesperidin supplementation in individuals with metabolic syndrome [41] and in overweight individuals [48].

Hesperetin, hesperidin, and its main metabolite have shown to boost mitochondrial energy production (spare capacity by 25%), to increase intracellular ATP by 33%, to reduce oxidative stress in a human skeletal muscle cell model [49]. Moreover, this same study also reported that the intake of hesperidin in aged mice (50 mg/kg/day) reverted the age-related muscle loss, improving its running performance. In the same direction, the effects of hesperidin on biochemical parameters and physical performance have been also studied in humans. Pittaluga et al. (2013) [50] investigated the effect of supplementation with a self-administrated amount of 250 mL of fresh red orange juice (ROJ) (natural source of hesperidin), thrice a day and 1 h before each meal during 4 weeks and after a single bout of exercise until exhaustion in healthy trained elderly women. The working capacity expressed as metabolic equivalents (METs) was significantly higher after ROJ supplementation (+9.0%) than in placebo (−1.5%), while there was no significant increase of maximal oxygen uptake (VO_2max_) in any group.

Previous studies have observed effects in biochemical markers after the intake of hesperidin, De Oliveira et al. (2013), after supplementation with hesperidin (glucosyl hesperidin; 100 mg/kg body mass) led to a decline of serum glucose with combined beneficial effect on swimming. Continuous or intermittent swimming with hesperidin supplementation lowered total cholesterol (−16%), low-density lipoprotein C (LDL-C) (−50%) and triglycerides (−19%), and increased high density lipoprotein (HDL-C) (48%) [51] in rats. Previous research has shown that daily consumption of 500 mL of orange juice for 3 months decreases lactic acidosis generated by the incremental exercise. The decrease in plasma lactate concentration was higher in the trained and hesperidin supplemented group (−27%) than in the control trained group (−17%) of overweight middle-aged women subjected to aerobic training [52].

Another ability of hesperidin is its capability to modulate the antioxidant state. For instance, De Oliveira et al. (2013) found that the consumption of hesperidin enhanced the antioxidant capacity on the continuous swimming group (183%) and decreased the lipid peroxidation (TBARS) on the interval swimming group (−45%) in rats [51]. Similarly, Estruel-Amades et al. (2019) observed an impact of hesperidin on the oxidant/antioxidant status of lymphoid tissues after an intensive training program was evaluated on rats. Supplementation with hesperidin, enriched in its 2S-isomer, led to a prevention of the increased ROS production and the decrease in superoxide dismutase (SOD) and catalase activities after an exhaustion test. These antioxidant effects of hesperidin were associated with a higher performance in the assessed training model [53].

In trained male athletes, supplementation for 4 weeks with 2S-Hesperidin (500 mg/day) improved cycling time-trial performance with a significant increase in power output during the exercise test [33]. Gelabert-Relato et al. (2019) compared acute and chronic effect (48 h and 15 days of supplementation) with high- and low-dose intake of polyphenols (luteolin and mangiferin), on sprint test and endurance exercise in physically active men [32]. The results showed a significant improvement in the sprint test (sprint 15 s after ischemia) in peak power output (PPO) and mean power output, after the polyphenol supplementation. Also, in the Wingate test, the experimental group improved by 4%. Also, Davis et al. (2010) conducted a crossover study examining quercetin’s ability to increase endurance capacity and maximal oxygen uptake (VO_2max_) in healthy untrained volunteers. They showed that a 7-day quercetin supplementation (1000 mg/day) produced improvements in time-to-fatigue and VO_2max_ by 13.2% and 3.9%, respectively, during a cycling test [17].

Previous studies, which have used different types of polyphenols and dosages, have observed enhancements in performance. Although the exercise protocols used were different, all of them showed improvements in exercise performance with high aerobic component [17,54,55,56,57]. However, to our knowledge, only one study reported improvements in the anaerobic component of exercise [32]. Thus, it is not clear if there is a beneficial effect of polyphenols on anaerobic exercise. Furthermore, only one study has examined the effect of hesperidin on exercise performance, showing an increase of absolute power output by 5% [33]. Moreover, given the importance of anaerobic component on performance in endurance sport and the limited evidence found related to the intake of hesperidin in anaerobic performance and its mechanisms, it seems necessary to conduct more research in order to clarify this relationship. 

Therefore, the aims of the present study were (1) to examine the effects of acute intake of Cardiose^®^ (500 mg of 2S-hesperidin) on anaerobic performance (peak power, power average, time to peak power, max speed and total energy (∑ 4 sprint test), (2) to determine the metabolic markers during exercise in ventilatory threshold 1 (VT1), and (3) to compare oxidative/antioxidant state during a rectangular test and after 24 h recovery in amateur cyclists. The recommendations from this experimental study will have the potential to inform about the optimal supplementation guidelines to optimise the performance and the recovery practices in athletes and provide sports nutritionist key information regarding the effects of polyphenol intake on sports performance and markers of oxidative stress. 

## 2. Methodology

Fifteen healthy male amateur cyclists (Table 1) completed the study. All participants had to meet the following inclusion criteria: aged 18–50 years, normal BMI (19–25 kg^.^m^−2^), at least 1-year of cycling experience, undergoing 6–12 h/week of training and being regular citrus consumers. The exclusion criteria were: (a) smokers or alcohol drinkers, (b) metabolic or cardiorespiratory pathologies or anomalies, (c) acute or chronic digestive pathologies that may interfere with the capacity to absorb nutrients, (d) injury in the last 6 months, (e) intake any type of supplementation or drug in the last 2 weeks, (f) no normal values in some parameter of the previous blood analysis and regular consumer of citrus fruits (≥5 oranges and derivatives/week). All subjects signed their informed consent before participating in the study. The study was performed in accordance with the guidelines of the Helsinki Declaration for Human Research [58] and was approved by the Ethics Committee of the Catholic University of Murcia.

### 2.1. Experimental Design

A randomized, single-blinded cross-over design was performed (Figure 1). Participants completed a total of 2 exercise sessions. Five hours before the exercise sessions, they ingested Cardiose^®^ (500 mg of 2S-hesperidin) or placebo (500 mg of microcellulose), supplied by HealthTech BioActives (Murcia, Spain). Cardiose^®^ contains standardized hesperidin (90% hesperidin, being at least 85% as 2S-Hesperidin isomer) from sweet orange (*Citrus Sinensis*). 

### 2.2. Procedures

Participants visited the Research Center for High Performance Sport at the Catholic University of Murcia at six different times. Visit 1 consisted of a medical examination, blood extraction, and anthropometry. Visit 2 consisted of a 24 h diet questionnaire before testing and an incremental test until exhaustion on a bike. Five hours prior to visit 3, participants were supplemented with Cardiose^®^ or placebo, according to the treatment arm. During Visit 3, participants underwent another 24 h diet questionnaire before testing, a 20 min test at ventilatory threshold 1 (VT1) intensity on a bike before and after a repeated all-out sprints test on a cycle ergometer, and four blood extractions. Visit 4 (24 h following visit 3) consisted of blood extraction and obtaining 24 h urine collection from the participant. Visits 5, 6, and 7 involved the same procedures performed as in visit 2, 3 and 4, respectively, but five hours prior to visit 6, participants were supplemented the other ingredient (Cardiose^®^ or placebo). There were no significant differences between the 24-h diet questionnaire made by the subjects. A standardized breakfast was consumed 2.5 h prior to each testing session (visits 2 and 3). The breakfast contained 95.16 gr of carbohydrates (68%), 18.86 g of protein (14%) and 11.30 g of lipids (18%), prescribed by a sports nutritionist. 

### 2.3. Tests 

#### 2.3.1. Medical Exam

The medical exam included a medical history, resting electrocardiogram, and medical examination (auscultation, blood pressure reading, etc.), so as to confirm that the participant was healthy and without any risk to be enrolled in the study.

#### 2.3.2. Anthropometry

The same researcher (International Society for the advancement of the KinanthropometryLevel-1 certified) performed the anthropometric measurements in both pre- and post-test. Height and body weight were measured using a digital scale for clinical use with a stadiometer (SECA 780; Vogel & Halke GmbH & Co., Hamburg, Germany). The skinfold thickness was assessed in accordance with ISAK guidelines [59], using Holtain Skinfold Calipers (Holtain Ltd., Crymych Pembrokeshire, UK). Percentage of body fat was determined with the Faulkner Equation [60] and the percentage of muscle mass with the modified Matiegka equation [61]. The sum of the eight skinfolds was also calculated.

#### 2.3.3. Maximal Test

An incremental step test was performed with the metabolic cart (Metalyzer 3B. Leipzig, Germany) to determine the ventilatory threshold 1 and 2 (VT1 and VT2) and VO_2max_. The test started at 35 W and increased 35 W every 2 min until exhaustion. To verify VO_2max_, the following criteria were assumed: plateau in the final VO_2_ values (increase ≤2.0 mL·kg^−1^·min^−1^ in the 2 last loads), maximal theoretical heart rate (HR) (220—age), respiratory exchange ratio (RER) ≥1.15 and a lactate value ≥8.0 mmol·L^−1^ [62,63]. The ventilatory threshold was obtained using the ventilatory equivalents method described by Wasserman [64].

#### 2.3.4. Rectangular Test

A constant effort was carried out on the bike at VT1 intensity during 20 min before and after the repeated all-out sprints test. The main objective of this test was to determine cardiorespiratory variables (VO_2_, CO_2_, RER, HR, and exercise economy) during a steady effort at low-intensity (Figure 2). This test was conducted before and after the repeated sprints test.

#### 2.3.5. Repeated Sprints Test

The exercise protocol consisted of 4 × 30 sec all-out sprints (Wingate test; WAnT) performed on a cycle ergometer (Monark Ergomedic 894E Peak Bike, Vansbro, Sweden) with 5 min of rest between sprints. For every sprint, the breaking resistance was constant (7.5% of body mass) and individualized for each participant (Figure 2). All subjects were verbally encouraged to continue to pedal as fast as they could for the entire 30 s. Peak power and anaerobic capacity were calculated and recorded in watts (W) and watts per kilogram body weight (W/kg^−1^). The total energy was calculated as the energy produced during the four test sprints in joules (J).

#### 2.3.6. Blood and Urine Analysis

Venous blood samples were taken for general analytics, in one tube of 3 mL ethylenediaminetetraacetic acid (EDTA) for hemogram, and in another tube of 3.5 mL with polyethene terephthalate (PET) for biochemical parameters. Red blood cells count (RBC) was carried out in an automated Cell-Dyn 3700 analyzer (Abbott Diagnostics, Lake Forest IL, USA), using internal (Cell-Dyn 22, Abbott Diagnostics, IL, USA) and external (Program of Excellence for Medical Laboratories-PEML) controls. Values of erythrocytes, hemoglobin, haematocrit, and hematimetric indexes (mean cell volume, MCV; mean cell haemoglobin, MCH; mean corpuscular hemoglobin concentration, MCHC; and red cell distribution width, RDW) were estimated.

Additionally, venous blood samples were collected pre VT1 test, post repeated sprint test, post second VT1 test, and 24 h after the end of the testing session for the measurement of antioxidant parameters (Figure 2). At each of the extraction points, 6 tubes of 3 mL of EDTA were obtained and one of them was centrifuged at 3500 rpm at 4 °C during 10 min and sent to the laboratory for later analysis. Urine samples, corresponding to 24 h urine collection from each participant after the supplementation, were frozen in liquid nitrogen after collection and thawed for its analysis.

#### 2.3.7. Hesperidin Metabolites Urine

Fifty µL of urine were mixed with 100 µL of water with 1% formic acid containing the internal standard (rac-Hesperetin-d3). Then, the mixture was injected into LC-MS/MS (UHPLC 1290 Infinity II Series coupled to a QqQ/MS 6490 Series Agilent Technologies, Sta. Clara, CA, USA). The method was validated using a pool of samples by determining the limit of detection (MDL) and quantification (MQL), repeatability (expressed as relative standard deviation RSD), and accuracy (%). Metabolites were quantified by external standard calibration using rac-Hesperetin-d3 as the internal standard. 

### 2.4. Markers of Oxidative Stress and Antioxidant Status

#### 2.4.1. TBARS

Thiobarbituric acid reactive substances (TBARS) are a by-product of the oxidative degradation of lipids by reactive oxygen species (lipid peroxidation), a commonly used marker of oxidative stress [65]. The principle of the method consists of isolating the lipid fraction of the plasma by precipitation of the lipids with phosphotungstic acid, followed by a reaction with thiobarbituric acid (TBA) that forms an adduct that allows detection by UV-VIS spectrophotometer at a wavelength of 532 nm.The assay involves the reaction of malondialdehyde (MDA), a product of lipid peroxidation, with thiobarbituric acid (TBA) under high temperature and acidic conditions to form an MDA–TBA complex that can be measured colorimetrically [66].

#### 2.4.2. Catalase

Catalase (CAT) activity was determined using a UV-VIS spectrophotometer. The principle of the method is that the absorbance of H_2_O_2_ decreases at 240 nm proportional to its decomposition, so that the concentration of H_2_O_2_ is critical in this determination. The decrease in absorbance per unit time is the measure of catalase activity. This is expressed in sec^−1^ per gram of hemoglobin [67]. The coefficient of variation between replicas must be less than or equal to 4.9%.

#### 2.4.3. SOD

Superoxide dismutase (SOD) activity was measured using a SD125 Ransod kit (Randox Ltd., Crumlin, United Kingdom). This method consists of the use of xanthine and xanthine oxidase to produce superoxide anion (O_2_^−^), which responds with the 2-(4-iodophenyl)-3-(4-nitrophenol)-5-phenyltetrazolium chloride (INT) reactive and forms a red complex detectable at 420 nm. The activity of SOD was measured through the inhibition of this reaction. The SOD activity is then quantified by measuring the degree of inhibition of this reaction [68]. The coefficient of variation between replicas must be less than or equal to 5.1%.

#### 2.4.4. Glutathion

Glutathion (GSH) was analyzed by the glutathione-S-transferase assay described by Akerboom and Sies [69]. The GSH was determined from whole blood, which was treated with perchloric acid to a final concentration of 6%, obtaining the supernatant after vortexing and centrifuging at 10.000 rpm for 10 min. After collecting the supernatants in vials, it was quantified by high performance liquid chromatography (HPLC) using a Waters ODS S5 NH2 Column (0.052, 25 cm) for separation purposes. Glutathion oxidized form, glutathione disulphide (GSSG), was determined in a similar way to GSH as shown above, as described by Asensi [70].

### 2.5. Statistical Analyses

The statistical analysis was performed using the Statistical Package for Social Sciences (SPSS 21.0, International Business Machines Chicago, IL, USA). Descriptive statistics are presented as mean ± SD. the assumption of normality was verified using the Shapiro–Wilks test. Paired sample t-test was used to evaluate differences between groups (Cardiose^®^ or placebo supplement). Moreover, a General Linear Model (repeated measures, analysis of variance, ANOVA) was performed for analyzing the within-group effects of the intake of the Cardiose^®^ or placebo supplement (4 time points). Additionally, the standardized mean differences (Cohen’s d (ES)) between groups was calculated together with the 95% confidence intervals and *η²* to analyze the size between groups. Finally, the relationships between levels of excreted hesperidin metabolites in urine and total energy and catalase activity were analyzed using Spearman correlation analysis (*r*). For all procedures, a level of *p* ≤ 0.05 was selected to indicate statistical significance.

## 3. Results

### 3.1. Repeated Sprint Test

Results for each variable were analyzed in two different ways: taking the best data of each of the four sprint test included in the series, and considering all the sprints as a unique exercise, using for each variable the average of all sprints results.

On one hand, taking into account the data corresponding to the best sprint, significant positive changes were observed in Cardiose^®^ compared to placebo in average power, maximum speed, and total energy (∑ 4 sprint test). However, no significant changes were found in peak power and time-to-peak power comparing Cardiose^®^ versus placebo group using these data (Table 2) (Figure 3). 

Considering the average values of the four sprints trials, positive changes were observed in the peak power, time-to-peak, and total energy in the Cardiose^®^ group compared to placebo, but not reaching the statistically significance (Table 2) (Figure 3). In addition, there was a positive significant correlation (*r* = 0.547; *p* = 0.043) between the levels of excreted hesperidin metabolites in urine and the difference in total energy (∑ 4 sprint test) between the placebo and supplemented group.

### 3.2. Metabolic Parameters

Metabolic parameters were evaluated and compared for the rectangular tests (20 min at VT1) carried out before and after the repeated all-out sprints test. During the VT1 testing period, no significant differences were observed in pre-post changes between Cardiose^®^ and placebo group (Table 3).

In relation to the cardiorespiratory parameters, a nonsignificant increase in oxygen consumption and a significant decrease in CO_2_ production were observed in both groups. However, no significant differences between groups were observed for these two parameters. In the same way, respiratory exchange ratio (RER), which is the ratio between the amount of CO_2_ produced and oxygen used in metabolism, significantly decreased from the first to the second rectangular test in both groups, but without significant differences between them. Exercise economy is defined as the oxygen uptake relative to body mass used at the VT1 workload. After the repeated sprint test, there was a non-significant increase in exercise economy in both groups. Also, there were no significant differences between groups. Moreover, the high-intensity repeated sprint test caused a significant increase in heart rate during the second VT1 stage of rectangular test in both groups. However, there were no relevant differences between supplements (Table 3).

Regarding energy substrates, a significant decrease in carbohydrates consumption and a significant increase in fat energy contribution from the first to the second rectangular test was observed in both groups, as expected after an intense physical activity such as repeated sprint test. Despite no significant differences observed between the Cardiose^®^ and placebo groups, this modulation of the energy substrates was slightly different in both supplements. Cardiose^®^ led to a higher increase (+15.1%) in the fat energy contribution (Figure 4).

### 3.3. Antioxidant Parameters

Effects of each treatment on the modulation of oxidative status were evaluated by the activity of the endogenous antioxidant system, such as SOD, CAT, GSH, GSSG, and TBARS, as biomarker of lipid peroxidation. These markers were evaluated prior to the testing session (E1: pre VT1 test, and 5 h after supplementation), post repeated sprint test (E2), post second VT1 test (E3), and 24 h after the end of the testing session (E4).

Regarding the activity of the CAT enzyme, the exercise protocol included in this study led to significant differences in CAT activity at the different evaluated points in both groups. For instance, a significant decrease was observed from the end of the repeated sprint test to the end of rectangular test in both groups. Regarding differences between the Cardiose^®^ and placebo groups, catalase activity was slightly increased in the Cardiose^®^ group after supplementation with this product—prior to the testing session, after the repeated sprint test, and at the end of the testing session—despite catalase activity being almost the same in both groups 24 h after the end of the testing session. However, these differences between groups were not statistically significant (Table 4). An inverse significant correlation between the levels of hesperidin metabolites excreted in urine and the percentage variations in the catalase activity at points E1 and E3 (*r* = −0.625; *p* = 0.013) between the placebo and supplemented group was observed.

Regarding the activity of SOD, repeated sprint test increased SOD activity both in Cardiose^®^ and placebo groups. However, a completely different trend was observed after this strenuous exercise in SOD activity of both groups. At the end of the rectangular test, SOD activity decreased in the Cardiose^®^ group, while it slightly increased in the placebo group. In contrast, both groups experienced a decrease in SOD activity from the end of the physical test to 24 h after exercise. However, these differences were not statistically significant (Table 4) (Figure 5).

Physical activity included in the exercise protocol increased TBARS levels in both groups. However, a greater attenuation of lipid peroxidation, identified by a decrease in TBARS, was observed in the Cardiose^®^ group from after repeated sprint test to the end of rectangular test. However, no significant changes were observed between interventions and between blood extraction points (Table 4) (Figure 5).

No significant decrease in GSH levels during the exercise protocol was observed in both groups. Despite levels of this antioxidant peptide being higher after Cardiose^®^ supplementation at baseline and after the repeated sprint test, differences between treatments were not significant (Figure 5). A positive significant correlation between the levels of hesperidin metabolites in urine and the percentage variations in the levels GSH %Δ 01-02 (*r* = 0.551; *p* = 0.033) between the placebo and supplemented group was observed.

A completely different trend in GSSG/GSH ratio was observed in both groups during the testing session. In the Cardiose^®^ group, GSSG/GSH ratio decreased during the testing session: after the repeated sprint test, and after the end of rectangular test. In the placebo group, GSSG/GSH ratio increased during the physical exercise: after the repeated sprint test and after the end of the testing session. Despite this different behavior in the GSSG/GSH ratio during the test, differences in the GSSG/GSH ratio were not statistically significant (Table 4) (Figure 5).

### 3.4. Hesperidin Metabolites Urine

Different hesperidin metabolites, mainly hesperetin glucuronides and sulfates, were analyzed in urine of the participants after the intake of Cardiose^®^. The main metabolite was Hesperetin-3-glucuronide, representing 78.9 ± 5.0% (*n* = 15) of the total, while hesperetin-7-glucuronide and hesperetin-7-sulfate were 6.9 ± 2.9% (*n* = 15) and 14.7 ± 4.1% (*n* = 15) of the excreted metabolites, respectively. Despite the similarities in the excreted metabolites profile, a large interindividual variability was observed in the amount of hesperidin metabolites excreted, ranging from 2.3 to 37.5 μmol. These wide differences between subjects in the absorption and excretion of hesperidin have been already reported [71].

## 4. Discussion

The main objective of this study was to assess the acute effects of 500 mg of 2S-hesperitin on physical performance, specifically in exercise with high anaerobic component [32], and secondary in metabolism and antioxidant status in amateur cyclists. The results showed that a single supplementation with 500 mg amount of 2s-Hesperidin may improve anaerobic parameters in a repeated sprint test for the Cardiose^®^ group. In addition, an improvement in antioxidant capacity and energy metabolism were observed after Cardiose^®^ supplementation during the exercise protocol.

A significant improvement in average power (+2.27%), maximum speed (+3.23%), and total energy (+2.64%) was observed after Cardiose^®^ supplementation when the best data of the sprint test series were considered. However, no significant improvements in anaerobic performance parameters were found for Cardiose^®^ group when average values of the repeated sprint test were evaluated. In addition, a positive correlation between excreted hesperidin metabolites in urine and the differences in total energy (Ʃ 4 test) between placebo and Cardiose^®^ was also found. Therefore, these data show that supplementation with Cardiose^®^ improves physical performance in an anaerobic trial such as the repeated sprint test. These results are in line with the improvement in physical performance observed in trained rats after 2S-Hesperidin supplementation [53] or in cycling time-trial performance in trained male athletes after supplementation with 2S-Hesperidin (500 mg/day) for 4 weeks [33]. These studies reported improvements of 58% in the time the exhaustion test and 5% in absolute power output a 10 min time-trial, respectively. Since anaerobic power is a key factor in sport performance [72], but sometimes difficult to improve, achieving small improvements in anaerobic performance as those described in this study may be very important for athletes, especially in high sports performance.

Antioxidant status and endogenous antioxidant capacity are key factors for the athlete’s performance [73]. Especially during high-intensity and short-duration or low-intensity and high-duration exercises which provoke high production of free radicals (ROS), these may be mediated through a variety of pathways [74]. Our study showed small changes in different antioxidant enzymes (CAT and SOD), peptides with antioxidant activity (GSSG/GSH), and oxidation markers (MDA-TBARS) between the Cardiose^®^ and placebo groups. Exercise-induced ROS production causes lipid peroxidation [31], superoxide anion generation through xanthine oxidase (XO) activation, and the increase in oxidized/reduced glutathione (GSSG/GSH) ratio [28,29]. Enzymes like SOD and glutathione are important antioxidant defences that protect cells from ROS-induced oxidative stress [75]. Oxidative stress may cause cellular damage through modifications to macromolecules, including proteins, lipids, and nucleic acids, and can occur as a result of high-intensity or moderate- to long-duration exercise [30]. In our study, the intense physical exercise causes an increase in CAT activity, which was observed in both experimental groups. An increase in the activity of CAT, versus placebo, was observed following the acute supplementation with Cardiose^®^. Also, an inverse correlation between the excreted levels of hesperidin metabolites in urine, and the percentage variations in the activity of CAT %Δ 01–03 (*p* = 0.013) was observed. These results suggest that the acute intake of Cardiose^®^ might promote the activity of this antioxidant enzyme. In rats submitted to intense exercise, 2S-hesperidin supplementation contributed to maintain catalase activity, and avoid changes induced by physical activity [53]. An increase in catalase activity during exercise may offer an advantage in high intensity efforts (e.g., sprint), where there is a high rate of ROS production, decreasing damage to the muscle cell.

Hesperidin has been also described to increase the activity of this antioxidant enzyme during senescence [76] or modulate its activity when it is impaired by different conditions [77,78]. In general, intense physical activity increases SOD activity [79]. However, a decrease in SOD activity after repeated sprint test to the end of rectangular test was observed in Cardiose^®^ (−5.9%) but not in placebo (+0.9%). This decrease was maintained 24 h after the end of exercise session. Cardiose^®^ seems to reduce the overexpression of SOD induced by physical exercise. In previous studies, supplementation with 2S-Hesperidin decreased SOD activity in trained rats [53]. Due to its scavenging activity hesperidin neutralizes reactive oxygen species—such as superoxide anion—generated during conditions of oxidative stress, as intense physical exercise. The decrease in SOD activity may be related to the reduced need for this endogenous enzyme when an exogenous antioxidant, such as hesperidin or other flavonoids, is provided [80,81]. Therefore, this decrease in SOD activation would indicate a lower production of free radicals, which leads to less damage to muscle cell structures and a better post-exercise recovery.

As we have already mentioned, intense physical activity increases ROS production and consequently lipid peroxidation, producing an increase of malondialdehyde and TBARS [82]. ROS produced during physical activity may react with unsaturated fatty acids comprising cellular membrane, leading to lipid peroxidation, a chain reaction that oxidizes fatty acids and produces more ROS [83]. In our exercise protocol, high-intensity exercise increased TBARS in both groups. However, a greater attenuation of lipid peroxidation (TBARS) was observed in Cardiose^®^ (−5.7 %) from after repeated sprint test to the end of rectangular test versus placebo (−2.3%). In previous studies, in rats subjected to interval swimming, the intake of hesperidin lowered (−45%) the lipid peroxidation [51]. Flavonoids such as hesperidin play a key role as free radical scavengers in vivo, preventing the increase in lipid peroxidation associated with high-intensity exercise. Furthermore, the antioxidant activity of citrus flavanones is not only related to their radical scavenging activity, but also to their ability to increase cellular defences via the Nrf2-ARE pathway, which regulates the expression of antioxidant genes including SOD, CAT, HO-1, GPX, and TXN, decreases intracellular pro-oxidants, and enhances antioxidant enzymes [76]. 

On the other hand, glutathione is a widespread peptide with antioxidant properties that may be found in plasma either as glutathion (GSH) or as glutathione disulfide (GSSG), its oxidized form [84]. Cardiose^®^ supplementation led to no significant increase in glutathione (GHS) levels, and additionally a significant correlation (GSH %Δ 01–02, *r* = 0.551; *p* = 0.033) between the levels of hesperidin metabolites in urine and the percentage variations in the GSH levels was observed. The ratio between the oxidized (GSSG) and reduced (GSH) glutathione form is also evaluated as an antioxidant status marker [85]. A different trend in this ratio was observed according to the supplementation. Placebo led to an increase in the GSSG/GSH ratio during the exercise protocol, while Cardiose^®^ supplementation decreased the GSSG/GSH ratio. These results suggest that Cardiose^®^ promotes glutathione antioxidant role, indicating a better antioxidant status in the experimental group. The reason of these modifications in GSH and GSSG could be caused by low concentrations of lipoperoxides and hydrogen peroxide, which are metabolized by glutathione peroxidase (GPX), generating an increase in GSSG [86]. This increase is neutralized by the increase in the activity of glutathione reductase (GR) [87]. This effect can explain the finding from our study with GSH and GSSG/GHS ratio. In previous works [88,89], hesperidin supplementation has been shown to minimize the impairment of glutathione antioxidant system induced by different alterations, restoring the usual levels of this body’s antioxidant peptide. All these changes in the endogenous antioxidant system (CAT, SOD, GSH, and GSSG) generate an ideal muscular environment to improve performance and recovery. Taken together, these results suggest that the intake of Cardiose^®^ affects the body’s own antioxidant capacity, even after an acute single intake. 

Improvements in endurance sports’ performance may be due to metabolic adaptations, which could be explained by the activation of PGC-1α (a key regulator of energy metabolism that increases biogenesis and mitochondrial working capacity). Finally, in terms of the cardiorespiratory parameters analyzed, no significant differences were found between the Cardiose^®^ group and placebo. Furthermore, no significant differences between treatments were found regarding the consumption of energy substrates, carbohydrates and fats, but Cardiose^®^ supplementation promoted the use of fats as energy substrates (+15.1% versus placebo). Polyphenols induce changes in PGC-1α activity via increased activation of the intracellular signalling pathways AMP-activated protein kinase (AMPK). Another factor that promotes metabolic adaptations induced by exercise is NRF2, a member of the Cap-N-Collar family of transcription factors, plays an important role in mitochondrial biogenesis, and variants of the NRF2 gene have been associated with endurance performance [90,91]. In vitro studies have shown that hesperidin also activates AMPK stimulating its phosphorylation [41]; besides this, in animal tissue an increase in NRF-2 expression was also observed [76]. These metabolic changes generate muscular level adaptations that prioritize the oxidation of fatty acids versus glucose, leading to higher energy efficiency [92,93,94,95,96], and therefore may predispose the athlete to a better sports performance. However, the small differences in oxygen consumption at the same intensity indicates a better exercise economy and, consequently, an improvement in performance.

As shown previously, Cardiose^®^ supplementation seems to improve physical performance during a complete exercise protocol, modulating athlete’s oxidative status during the physical activity in semi-professional cyclists. The absence of human studies with hesperidin including anaerobic power tests, makes difficult any comparison. It is important to highlight that these results were obtained after an acute and single intake of 500 mg of Cardiose^®^ and placebo. The small changes observed after this single intake may be increased after a chronic consumption of this product. The main limitation of this study was the size of the sample. A larger sample could improve our results´ power and the lack of previous research studies on the topic. Future research should be conducted to evaluate the chronic effect of 2S-hesperidin supplementation on sports performance and oxidative stress, as well as to clarify if hesperidin can improve physical performance during high-intensity exercise.

## 5. Conclusions

A single acute intake of Cardiose^®^ (500 mg of 2S-hesperidin) improves performance in maximum anaerobic effort in semi-professional cyclists. In addition, oxidative status and antioxidant defenses were slightly modulated. These findings could help improve performance in high-intensity exercises for both amateur and high-performance athletes.

## Figures and Tables

**Figure 1 nutrients-11-01898-f001:**
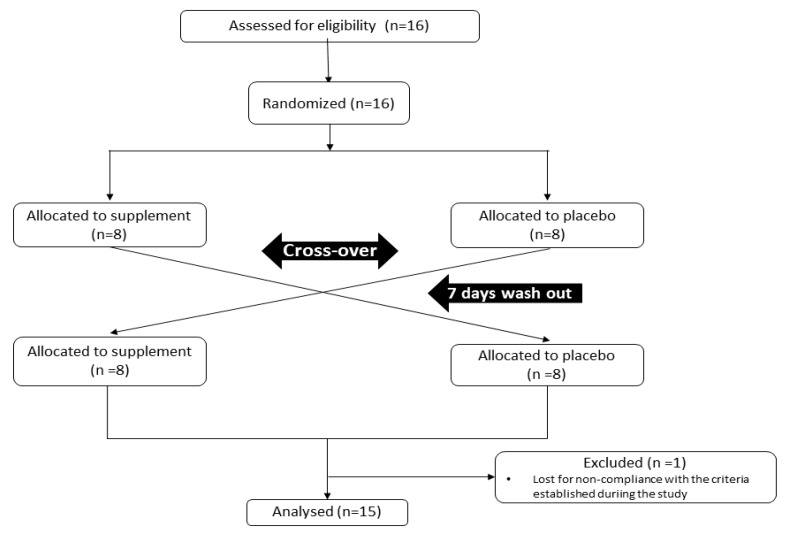
Consolidated Standards of Reporting Trials flow chart of participants during the study intervention.

**Figure 2 nutrients-11-01898-f002:**
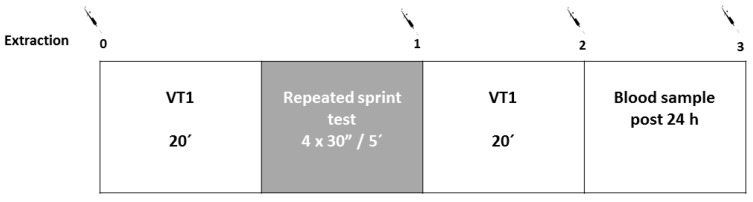
Exercise protocol and blood sampling plan during Visit 3/6. VT1 = ventilatory threshold 1; syringe = blood sample.

**Figure 3 nutrients-11-01898-f003:**
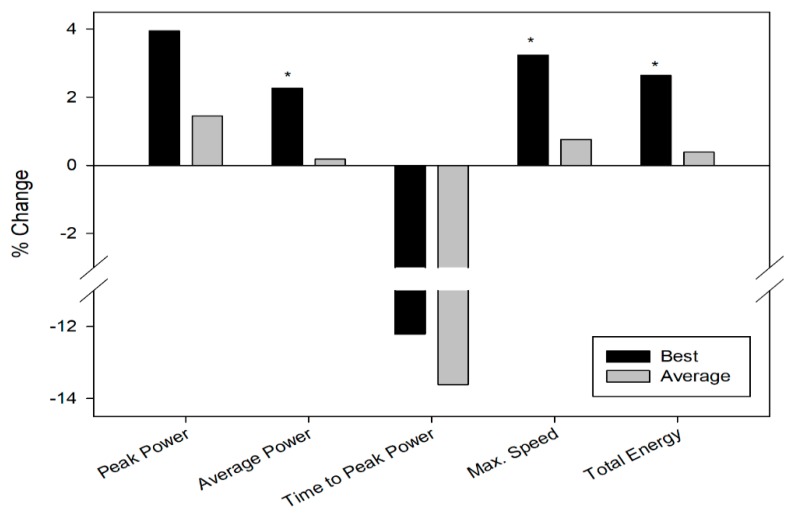
Changes in the repeated sprint test results after supplementation with Cardiose^®^ using best data of each of the four sprint tests and the average of all sprints. * = between-group significant changes (*p* < 0.05).

**Figure 4 nutrients-11-01898-f004:**
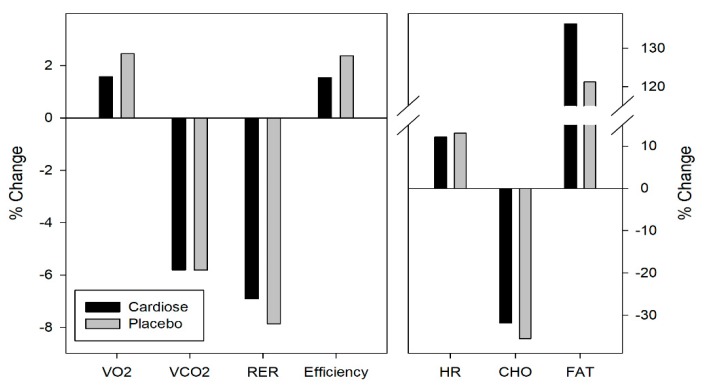
Changes in cardiorespiratory and metabolic parameters between the rectangular test (VT1 intensity during 20 min) carried out before and after the repeated all-out sprints test.

**Figure 5 nutrients-11-01898-f005:**
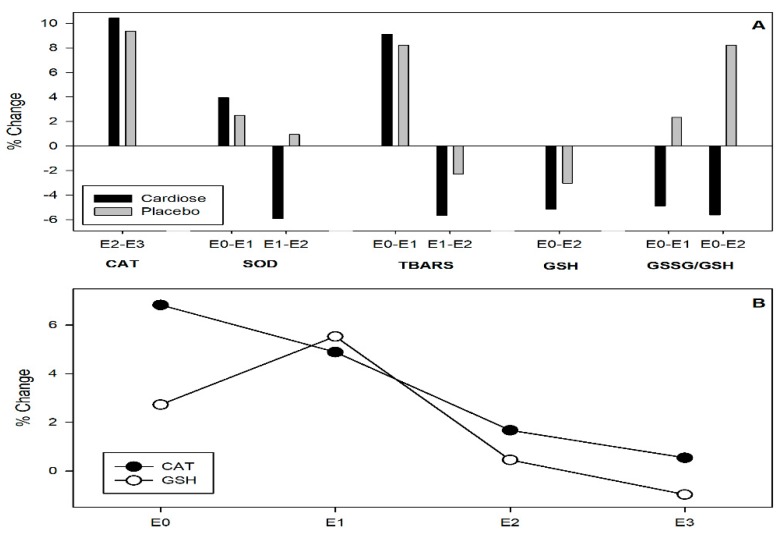
Changes in antioxidants and TBARS measured in repeated sprint test. (**A**) Changes between different time points. (**B**) Changes between treatments. Antioxidant markers were evaluated prior to the testing session (pre VT1 test, and 5 h after supplementation) [E0], post repeated sprint test [E1], post second VT1 test [E2], and 24 h after the end of the testing session [E3].

**Table 1 nutrients-11-01898-t001:** Baseline general characteristics of the study participants.

Age (years)	Height (cm)	Weight (kg)	BMI (kg/m^2^)	BF (%)	VO_2max_ (mL·kg^−1^·min^−1^)	VT1 (%)	VT2 (%)
33.3 ± 7.9	174.9 ± 4.2	69.4 ± 4.5	22.7 ± 1.2	11.2 ± 2.2	61.6 ± 7.4	53.0 ± 6.1	86.0 ± 4.7

BMI = Body mass index; BF = Body fat; VO_2max_ = Maximum oxygen consumption; VT1 = Ventilatory threshold 1 (aerobic); VT2 = Ventilatory threshold 2 (anaerobic).

**Table 2 nutrients-11-01898-t002:** Repeated sprint test outcomes.

Parameters	Best Sprint Data		Average (All Sprints)	
	Cardiose^®^	Placebo	Cardiose^®^	Placebo
	Mean ± SD	Mean ± SD	Mean ± SD	Mean ± SD
**PeakPower (w)**	835.50 ± 96.08	803.79 ± 110.43	740.16 ± 74.52	729.55 ± 91.36
**Poweraverage (w)**	567.84 ± 55.44 *	555.25 ± 51.81 *	511.71 ± 52.68	510.78 ± 52.99
**Time to peakpower (ms)**	2840.69 ± 715.99	3235.85 ± 1516.06	3003.13 ± 950.28	3476.14 ± 1546.57
**Max speed(rpm)**	132.86 ± 9.59 *	128.70 ± 9.24 *	120.83 ± 7.79	119.92 ± 9.79
**Totalenergy (J)**	16246.29 ± 1600.37 *	15827.79 ± 1505.86 *	14874.79 ± 1570.83	14818.36 ± 1608.24

Δ = percentage of pre-post change; W = watts; ms = millisecond; J = joules; Max speed = maximum speed; rpm = revolutions per minute SD = standard deviation. * = between-group significant changes (*p* < 0.05).

**Table 3 nutrients-11-01898-t003:** Metabolic parameters in VT1 stage before and after the repeated sprint test.

Metabolic parameters		Cardiose^®^	Placebo
		Mean ± SD	Mean ± SD
VO_2_ (L/min)	Pre	2.11 ± 0.39	2.06 ± 0.39
	Post	2.15 ± 0.45	2.13 ± 0.51
VCO_2_ (L/min)	Pre	1.93 ± 0.41	1.86 ± 0.35
	Post	1.83 ± 0.43	1.77 ± 0.43
RER	Pre	0.91 ± 0.03	0.90 ± 0.02
	Post	0.85 ± 0.03	0.83 ± 0.02
Efficiency (mL/Kg/W)	Pre	3.97 ± 0.48	3.85 ± 0.39
	Post	4.02 ± 0.49	3.94 ± 0.51
HR (pul/min)	Pre	128.55 ± 9.53	128.04 ± 8.95
	Post	144.15 ± 12.50	144.74 ± 11.31
Carbohydrates	Pre	105.5 ± 33.3	97.6 ± 19.7
	Post	72.9 ± 28.5	63.6 ± 17.7
Fat	Pre	14.1 ± 6.0	15.9 ± 6.1
	Post	28.5 ± 4.0	31.8 ± 8.7

Δ = percentage of pre-post change; heart rate (beats·min^-1^); SD = standard deviation; VO_2_ = oxygen uptake (L·min^−1^); VCO_2_ = carbon dioxide production (L·min^−1^); W = watts; HR = Heart rate; RER = Respiratory exchange ratio.

**Table 4 nutrients-11-01898-t004:** Antioxidants and TBARS measured in repeated sprint test. Antioxidant markers were evaluated prior to the testing session (pre VT1 test, and 5 h after supplementation) [E0], post repeated sprint test [E1], post second VT1 test [E2], and 24 h after the end of the testing session [E3].

Antioxidant/Oxidant Status Markers	Cardiose^®^ Mean ± SD				Placebo Mean ± SD			
	E0	E1	E2	E3	E0	E1	E2	E3
CAT (U/g Hb)	25.66 ± 4.74	53.93 ± 13.41 *	27.53 ± 6.54 *	24.66 ± 4.27	24.02 ± 3.13	51.41 ± 16.41 *	27.07 ± 4.63 *	24.53 ± 4.18
SOD (U/g Hb)	1298.00 ± 261.75	1349.13 ± 225.31	1269.27 ± 271.13	1228.33 ± 229.77	1319.00 ± 145.54	1352.13 ± 201.31	1364.80 ± 272.74	1337.67 ± 193.97
GSH (nmol/mg protein)	25.02 ± 2.80	24.89 ± 2.90	23.73 ± 2.10	24.36 ± 2.75	24.36 ± 2.24	23.59 ± 3.37	23.62 ± 3.19	24.60 ± 1.72
GSSG (nmol/mg protein)	0.351 ± 0.073	0.334 ± 0.075	0.315 ± 0.067	0.378 ± 0.152	0.325 ± 0.073	0.316 ± 0.078	0.336 ± 0.068	0.388 ± 0.130
% GSSG/GSH	1.42 ± 0.32	1.35 ± 0.31	1.34 ± 0.31	1.54 ± 0.54	1.34 ± 0.28	1.37 ± 0.41	1.45 ± 0.37	1.57 ± 0.48
TBARS (nmol/mg protein)	2.49 ± 0.34	2.71 ± 0.45	2.56 ± 0.44	2.63 ± 0.26	2.43 ± 0.22	2.63 ± 0.36	2.57 ± 0.38	2.58 ± 0.32

Abbreviations: CAT = catalase; SOD = superoxide dismutase; GSH = Reduced glutathione; GSSG = oxidized glutathione; % GSSG/GSH = % oxidized glutathione/ Reduced glutathione; TBARS = Thiobarbituric acid reactive substances; SD = standard deviation. * = intra-group significant changes (*p* < 0.05)

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
