# Peer review of "Acute Effects of Hesperidin in Oxidant/Antioxidant State Markers and Performance in Amateur Cyclists"

_nutrients, 2019, doi:10.3390/nu11081898_

Round 1
Reviewer 1 Report
This is a well written paper with interesting conclusions. The study sample is small and followed up for a short period of time. The study is a pilot study and would be worth following it up with a longterm follow up (few weeks/months) to assess the impact of hesperidin on their performance.
I wonder why amateur cyclists and not professional cyclists were recruited to the study. Why only male cyclists were selected? Any particular reason?
The methodology mentions they completed questionnaires, but it does not say what they included. Drug history may be relevant in view of potential drug interaction with hesperidin? It is worth adding it to the methodology section. In terms of smoking status and alcohol consumption should be stated as it may affect oxidative function.
Author Response
Point 1:This is a well written paper with interesting conclusions. The study sample is small and followed up for a short period of time. The study is a pilot study and would be worth following it up with a longterm follow up (few weeks/months) to assess the impact of hesperidin on their performance.
Response 1: We thank to the reviewer for their constructive and helpful feedback on our manuscript. We have replied to each specific comment in the section below, and have marked the corresponding edits within the manuscript using tracked changes.
We also thought the results were interesting, so a 2-month study with 40 subjects and two intervention groups (hesperidin and placebo) was subsequently conducted, where were evaluated markers of antioxidant and inflammatory status, biochemical, metabolic and performance aerobic-anaerobic.
Point 2: I wonder why amateur cyclists and not professional cyclists were recruited to the study. Why only male cyclists were selected? Any particular reason?
Response 2: Professional cyclists were not included because, in our region, we do not have many athletes of that profile. Another reason is that cyclists would have had varying levels of fitness depending on their goals in the season at the time of data collection and that could cause have biased the results of the study. In addition, it was easier for us to control the training variables in amateur cyclists.
Men were only included in this study due to the limited budget for the study and considering that the blood tests for antioxidant markers were very expensive. In addition, our initial proposal was to include more male subjects, but the higher sample size was economically unviable.
Point 3: The methodology mentions they completed questionnaires, but it does not say what they included. Drug history may be relevant in view of potential drug interaction with hesperidin? It is worth adding it to the methodology section. In terms of smoking status and alcohol consumption should be stated as it may affect oxidative function.
Response 3: We performed a 24-hour food recall prior to the tests on days 2 and 3 to see if there were any changes in their dietary intake between subjects and tests. In order to minimize these potential differences, we included a standardized breakfast for all subjects before the tests on days 2 and 3, and the data for this meal are included in the methodology. A sentence has been included to state that not significant changes were found.
To our knowledge, hesperidin does not interact with any medication or drug. It is used as a food ingredient in the European Union (1). The interaction of hesperidin with drugs is very weak, as demonstrated by the scientific literature (2). What has been demonstrated is an interaction of grapefruit juice with certain drugs due to containing naringin (3) and furocoumarins (4). Hesperidin levels are low relative to other flavonoids, and we believe that the interaction with certain medications is due to the combined action of certain flavonoids and not just one.
As you have well commented, people who smoke or drink alcohol can modify their antioxidant status. Thus, in our study, this was a criterion of exclusion so that smokers and alcohol drinkers were not included in the study. However, we forgot to include this information and it will be included in the revised version.
EU Novel food catalogue (http://ec.europa.eu/food/safety/novel_food/catalogue/search/public/index.cfm#). Johnson, E. J., Won, C. S., Köck, K., & Paine, M. F. (2017). Prioritizing pharmacokinetic drug interaction precipitants in natural products: application to OATP inhibitors in grapefruit juice. Biopharmaceutics & drug disposition, 38(3), 251-259. Seden, K., Dickinson, L., Khoo, S., & Back, D. (2010). Grapefruit-drug interactions. Drugs, 70(18), 2373-2407. Bailey, D. G., Dresser, G. K., Kreeft, J. H., Munoz, C., Freeman, D. J., & Bend, J. R. (2000). Grapefruit‐felodipine interaction: effect of unprocessed fruit and probable active ingredients. Clinical Pharmacology & Therapeutics, 68(5), 468-477.
Reviewer 2 Report
The paper addressed an interesting subject concerning supplementation of Hesperidin on young cyclists before exercise. However, the language as well as the quality of presentation are poor. The text needs extensive editing.
Abstract
15 state
20 power, data
21 total energy? What does it mean? Total energy produced during exercise?
Introduction
It needs scientific rewriting and English check
36 fatty acids
37-40 consider rephrasing
54 no reference
56 too many repetitions of the word polyphenols
57 elimination of "the”
76 Src? Akt?
79 “a type of flavonoids including Hesperidin” this information should have been reported prior
85-89 check grammar and syntax
91 and ref 51 rewrite sentence and refer to changes in comparison to control group
101-103 consider rephrasing. Also “which is a natural source of hesperidin” is repeated
105 of overweight
107 consider rephrasing
108 of hesperidin
123 in?
126 improvements
128 that used
129 consider rephrasing
131 To our knowledge
128-137 This is a key paragraph leading to the aims of the study and has to be written more scientifically and comprehensively
138-146 The aims again have to be very specific
Methods
There are some methodological issues
153 undergoing
168 consider rephrasing
174-175 According to Fig 2 there are 2 x VT1 tests which are not stated. Also 4 blood collection time points are given, while here states 3
180 Time points of blood collection have to be very clearly stated. This is a major point for consideration throughout the text and especially when it comes to results
182 Tests
When medical exam and anthropometric measurements were done?
197 check elsewhere “ventilatory”
217 analysis
Did pre-VT1 sample served as baseline?
265 The
271 groups
272 not only these correlations were done
Results
Needs changes. There are some inconsistencies as well.
187 tests….or considering…
291 total energy variable has to be explained in methods. What does it mean? Total energy expended during exercise?
Table 2 Best sprint data
Figure 3 not needed
310 a non significant
315 defined as
316 consider rephrasing
318 “The intense physical activity of the exercise” needs scientific writing
318 heart
319 test
328 consider rephrasing of the title in Table 3
Figure 4 not needed
337 now it is reported that there was no blood sampling before VT2, while in the figure is shown no blood sampling after VT1. This is very confusing
342 again blood sampling points
354 grammar/ syntax mistakes
360 what between extraction times means?
363 baseline levels are not stated for both groups. It will help the text if it was clear whether changes were sig a) within groups, b) between groups
Table 4 Table has different indication of time points in comparison to the text
Figure 5 not needed. It is very confusing or wrong. There could be one figure on changes on all parameters during the 4 time points.
Discussion
Needs rewriting.
395 exercise
394-399 The conclusion has to be rewritten. This is not the specific conclusion of the study
Eg. A single supplementation with x amount of Hesperidin…may improve anaerobic parameters…..
Ref.53, 33 need to be elaborated and a comparison to be made with current findings
415 comma needed
415-477 this is a huge paragraph and difficult to read. One paragraph for each metabolite could be better.
416 grammar/ syntax mistakes
420 positive correlation
426 through
438 clinical significance of the findings here and elsewhere is not discussed
449 increased
451 subjected
468-470 consider rephrasing
It is recommended that you focus on significance or non significance
478 grammar/ syntax mistakes
493 eliminate to
495-496 what is the meaning?
497-516 Again specific conclusion needs to be given. Eg oxidative profile was not changed. Also NO measurement and lack of previous studies are not considered limitations of the study
Author Response
The paper addressed an interesting subject concerning supplementation of Hesperidin on young cyclists before exercise. However, the language as well as the quality of presentation are poor. The text needs extensive editing.
We thank to the reviewer for their constructive and helpful feedback on our manuscript. We have replied to each specific comment in the section below, and have marked the corresponding edits within the manuscript using tracked changes.
Abstract
15 state Changed
20 power, data: Corrected
21 total energy? What does it mean? Total energy produced during exercise?
This term refers to the total energy during repeated sprint exercise. Added to the manuscript.
Introduction
It needs scientific rewriting and English check
36 fatty acids: Corrected
37-40 consider rephrasing: We have considered your comment and we have rephrased the sentence in order to clarify its meaning.
54 no reference: Following your suggestion, we have included a reference here.
56 too many repetitions of the word polyphenols: Changed
57 elimination of "the”: Corrected
76 Src? Akt? Introduced the definition
79 “a type of flavonoids including Hesperidin” this information should have been reported prior Following your suggestion, we have included this information earlier (line 44).
85-89 check grammar and syntax: The grammar and syntax have been checked, as recommended.
91 and ref 51 rewrite sentence and refer to changes in comparison to control group: Following your suggestion, we have rewritten the sentence and included the percentage of change compared to the control group.
101-103 consider rephrasing. Also “which is a natural source of hesperidin” is repeated: Corrected
105 of overweight: Changed
107 consider rephrasing: We have rephrased, as recommended.
108 of hesperidin: Changed
123 in?: Removed
126 improvements: Changed
128 that used: Changed
129 consider rephrasing: Rephrased
131 To our knowledge: Changed
128-137 This is a key paragraph leading to the aims of the study and has to be written more scientifically and comprehensively: Following your suggestions, we have modified the paragraph in order to lead better to the aims
138-146 The aims again have to be very specific: Following your suggestion, we have modified the aims.
Methods
There are some methodological issues
153 undergoing: Changed
168 consider rephrasing: Changed
174-175 According to Fig 2 there are 2 x VT1 tests which are not stated. Also 4 blood collection time points are given, while here states 3: Changed
180 Time points of blood collection have to be very clearly stated. This is a major point for consideration throughout the text and especially when it comes to results. Corrected.
182 Tests: Changed
When medical exam and anthropometric measurements were done?
As indicated in line 170-171, medical examination and anthropometry were performed at visit 1 at the same time of day for all conditions.
197 check elsewhere “ventilatory”: Changed
217 analysis: Changed
Did pre-VT1 sample served as baseline? Yes, the pre-VT1 sample was used as baseline.
265 The: Changed
271 groups: Changed
272 not only these correlations were done
We carried out all possible correlations between hesperidin metabolites excreted in urine and the rest of measured variables. However, we only include those correlations that were significant and could explain our results.
Results
Needs changes. There are some inconsistencies as well.
187 tests….or considering… Corrected
291 total energy variable has to be explained in methods. What does it mean? Total energy expended during exercise? The repeated sprint test was composed of four 30s all-out efforts with 5min of recovery between sprints. The total energy is the sum of the energy produced in all 4 efforts. This information has now been included in methods. Amended
Table 2 Best sprint data: Changed
Figure 3 not needed: We appreciate your comment but we think that Figure 3 show more visual representation of our results.
310 a non significant: Changed
315 defined as: Corrected
316 consider rephrasing: Corrected
318 “The intense physical activity of the exercise” needs scientific writing: Changed
318 heart: Changed
319 test: Changed
328 consider rephrasing of the title in Table 3: Changed
Figure 4 not needed : We appreciate your comment but we think that Figure 3 shows a more visual representation of our results.
337 now it is reported that there was no blood sampling before VT2, while in the figure is shown no blood sampling after VT1. This is very confusing: We are not sure if we understood your point. However, as we have represented in Figure 2, there was no exercise in VT2 and the blood extractions were taken before starting the test, after the repeated sprint test, at the end of the second VT1 stage and 24h after the end of the exercise protocol.
342 again blood sampling points: These are the same blood extraction points that were detailed in lines 338 to 340. We believe that this type of nomenclature would make the text less dense and repetitive.
354 grammar/ syntax mistakes: Changed.
360 what between extraction times means? We have now specified in the main text that this means blood extraction times. Changed
363 baseline levels are not stated for both groups. It will help the text if it was clear whether changes were sig a) within groups, b) between groups: We have presented the baseline data in table 4.
Table 4 Table has different indication of time points in comparison to the text. We have re-checked this and the time points are the same in the main text as in table 4. If you need additional clarification, please let us know.
Figure 5 not needed. It is very confusing or wrong. There could be one figure on changes on all parameters during the 4 time points. Following your suggestion, we have modified this figure .
Discussion
Needs rewriting.
395 exercise: Corrected.
394-399 The conclusion has to be rewritten. This is not the specific conclusion of the study
Eg. A single supplementation with x amount of Hesperidin…may improve anaerobic parameters….. Changed
Ref.53, 33 need to be elaborated and a comparison to be made with current findings Changed
415 comma needed. Corrected
415-477 this is a huge paragraph and difficult to read. One paragraph for each metabolite could be better. Following your suggestion, we have divided the text into different paragraphs for each metabolite.
416 grammar/ syntax mistakes Changed
420 positive correlation: Corrected
426 through: Corrected
438 clinical significance of the findings here and elsewhere is not discussed: Amended
449 increased Corrected
451 subjected Corrected
468-470 consider rephrasing Modified
It is recommended that you focus on significance or non significance: We have considered your comment
478 grammar/ syntax mistakes: Corrected
493 eliminate to: Corrected
495-496 what is the meaning? This means that a lower oxygen consumption at the same intensity indicates a better exercise economy and, consequently, an improvement in performance . This has been clarified in the text. Response
497-516 Again specific conclusion needs to be given. Eg oxidative profile was not changed. Also NO measurement and lack of previous studies are not considered limitations of the study Changed

Reviewer 3 Report
In the Introduction the Authors should properly present the comparison between concentrated product and natural sources of hesperidin. The list of literature presented is not sufficiently clear. In Methods they should explain why they did not use a comparison with natural product and also if there are published comparisons on that. Moreover, in Introduction and in Discussion the Authors should divide the data observed in single bout exercise and the data obtained in long-term exercise. The Authors should define why they recruited cyclists, who are mainly aerobic athletes, and not some other typically anaerobic, e.g. rowers.
Pag 3 of 22 explain the term “international”
Pag 6 of 22 define method for hemogram, define imprecision for all methods
Discussion should shortened, avoiding the repetition of data presented in results
Author Response
We thank the reviewer for his/her constructive and helpful feedback on our manuscript. We have answered to each specific comment hereafter. We have highlighted each section of the manuscript that has been edited.
Ponit 1: In the Introduction the Authors should properly present the comparison between concentrated product and natural sources of hesperidin.
Response 1: Our purpose in the introduction was not to compare concentrated products with the natural source of hesperidin since the experimental product is almost 100% hesperidin. We have scoured the main natural sources of hesperidin and the differences between our molecule and others that are marketed; indicating that the natural form of hesperidin (2s-hesperidin) is conserved in a high proportion compared to the others. If you consider that we should introduce any specific change, we would thank if you could specify these changes.
Point 2: The list of literature presented is not sufficiently clear.
Response 2: Sorry, but we do not understand this comment. Could you be more specific with this question? We will be pleased to answer any comment about it.
Point 3: In Methods they should explain why they did not use a comparison with natural product and also if there are published comparisons on that.
Response 3: Unfortunately, we had a limited budget for this study and considering that the antioxidant markers blood tests were expensive, we decided to include only 2 groups (hesperidin and placebo). The initial idea was to introduce a group that would take a natural source of hesperidin but, in addition to the economic limitation, we also thought that these high-sugars foods might bias the study results. It is important to take into account that it was necessary to take at least 1-1.5 litres of orange juice to obtain the amount of 500 mg of hesperidin.
Point 4: Moreover, in Introduction and in Discussion the Authors should divide the data observed in single bout exercise and the data obtained in long-term exercise.
Response 4: Thanks for your suggestion. However, we have structured the introduction and discussion by physiological variables because we think that this would be easier for reader comprehension.
Point 5: The Authors should define why they recruited cyclists, who are mainly aerobic athletes, and not some other typically anaerobic, e.g. rowers.
Response 5: As you comment, cyclists mainly use their aerobic metabolism during competition. However, during the key moments of the competition, the anaerobic metabolism predominates. Therefore, if this kind of athletes improves their anaerobic component, they are likely to improve their performance. In addition, metabolic markers during a predominantly aerobic zone (ventilatory threshold 1) were also analysed in the study in order to find out if hesperidin improves fatty acids oxidation or modifies oxygen consumption, as other polyphenols have already shown. Moreover, as it is well known, cyclists have a more developed aerobic metabolism than other athletes that present a more anaerobic predominantly profile.
Pag 3 of 22 explain the term “international”:
This is a mistake. Sorry. The right term is interventional. We have changed it.
Pag 6 of 22 define method for hemogram, define imprecision for all methods
Following your suggestion, we have included the error of the measurement for the hemogram and the rest of the used methods in the methods section. That is what we have understood by reading this comment.
Point 6: Discussion should shortened, avoiding the repetition of data presented in results:
Response 6: We have revised the discussion section in order to minimise the repetition of the results.
Round 2
Reviewer 2 Report
Changes are made as suggested.